# Molecular imaging of humain hair with MeV-SIMS: A case study of cocaine detection and distribution in the hair of a cocaine user

Luka Jeromel[1], Nina Ogrinc[2¤], Zdravko Siketić[3], Primož Vavpetič[1], Zdravko Rupnik[1], Klemen Bučar[1], Boštjan Jenčič[1], Mitja Kelemen[1], Matjaž Vencelj[1], Katarina Vogel-Mikuš[1,4], Janez Kovač[1], Ron M. A. Heeren[2], Bryn Flinders[2], Eva Cuypers[2,5], Žiga Barba[1]*, Primož Pelicon[1]

1 Jožef Stefan Institute, SI-Ljubljana, Slovenia, 2 The Maastricht MultiModal Molecular Imaging Institute, Maastricht University, ER Maastricht, Maastricht, The Netherlands, 3 Rudjer Bošković Institue, Zagreb, Croatia, 4 Department of Biology, Biotechnical Faculty, University of Ljubljana, Ljubljana, Slovenia, 5 KU Leuven Toxicology & Pharmacology, Leuven, Belgium

¤Current address: Univ. Lille, Inserm, CHU Lille, U1192 - Protéomique Réponse Inflammatoire Spectrométrie de Masse – PRISM, Lille, France

* ziga.barba@ijs.si

**Data Availability Statement:** We have uploaded the relevant datasets to Zenodo with the DOI as follows: (10.5281/zenodo.6089948).

## Abstract

Human hair absorbs numerous biomolecules from the body during its growth. This can act as a fingerprint to determine substance intake of an individual, which can be useful in forensic studies. The cocaine concentration profile along the growth axis of hair indicates the time evolution of the metabolic incorporation of cocaine usage. It could be either assessed by chemical extraction and further analysis of hair bundels, or by direct single hair fibre analysis with mass spectroscopy imaging (MSI). Within this work, we analyzed the cocaine distribution in individual hair samples using MeV-SIMS. Unlike conventional surface analysis methods, we demonstrate high yields of nonfragmented molecular ions from the surface of biological materials, resulting in high chemical sensitivity and non-destructive characterisation. Hair samples were prepared by longitudinally cutting along the axis of growth, leaving half-cylindrical shape to access the interior structure of the hair by the probing ion beam, and attached to the silicon wafer. A focused 5.8 MeV $^{35}Cl^{6+}$ beam was scanned across the intact, chemically pristine hair structure. A non-fragmented protonated $[M+H]^+$ cocaine molecular peak at $m/z = 304$ was detected and localized along the cross-section of the hair. Its intensity exhibits strong fluctuations along the direction of the hair's growth, with pronounced peaks as narrow as 50 micrometres, corresponding to a metabolic incorporation time of approx. three hours.

## Introduction

The presence of illegal chemical substances in human body is usually monitored by bulk chemical analysis, or by spectroscopic methods of biological specimens [1–5], such as urine, saliva, sweat, and hair. Drugs and their metabolites remain chemically stable in hair over longer

**Funding:** Work at JSI was supported by the Slovenian research agency grants No. P1-0112, I0-0005, J7-9398 and N1-0090. Additionally, resources and within the EU H2020 project No. 824096 "RADIATE " and IAEA CRP projects F11019 "Development of Molecular Concentration Mapping Techniques Using MeV Focused Ion Beams" and F11021 "Enhancing Nuclear Analytical Techniques to Meet the Needs of Forensic Science". N.O. acknowledges funding by the European Union, European Social Fund, and the support from FP7 European Union Marie Curie IAPP Program, BRAINPATH. A traveling grant for a long stay abroad of E.C was awarded by Fonds Wetenschappelijk Onderzoek (FWO). The funders had no role in study design, data collection and analysis, decision to publish, or preparation of the manuscript.

periods of time, whereas the traces of drugs in urine or other body fluids typically wash out over a period of few days. The analysis of drugs and pharmaceutical compounds in human hair enables an insight into the recent history of (ab)use, recorded in a hair structure by the incorporation of molecules and their metabolic fragments [6–8]. Various pharmaceutical substances, as well as illegal substances were detected in hair [9, 10] with a variety of analytical techniques. Illegal substances detected in hair include cocaine [11], heroin [12], morphine [13], Δ9-tetrahydrocannabinol (THC) [14], codeine [15], amphetamines [16] and anabolic steroids [17] as well as prescriptive medication [18] and cosmetic products [19]. Mass spectrometry of human hair samples has even proven to be a useful tool to monitor patient's adherence to a drug treatment protocol [20].

Drugs or chemicals may enter the hair's structure by passive diffusion from the blood capillaries into the growing cells at the base of the hair follicle. With some time delay, the drugs may enter also into keratogenous zone during the formation of the hair shaft. In addition, drugs may enter by diffusion from sweat or sebum secretions [21]. The rate of drug incorporation into the hair depends on the melanin content of the hair, the lipophilicity and the basicity of the drug substance [22] as well as the growth phase of the hair. It is of great interest to know the scale and the intensity of the concentration fluctuation on a short time scale, i.e., daily or hourly, as this may reflect the short-time variations in the body liquid concentrations, revealing the frequency and the dosage of exposure, as well as the response of metabolism after the drug uptake.

Typical cocaine concentration values in hair samples after its administration have been reported from 0.5 to 216 ng/mg [23]. Chemical extraction is widely used to detect the presence of cocaine in hair, applying either standard acid hydrolysis, extraction with organic solvents or advanced methods of extraction, i.e. supercritical fluid extraction, followed by gas chromatography/mass spectrometry [24, 25]. By consecutive extractions of hair strand segments, typically 1 cm long, long-term records of (ab)use may be obtained. The fluctuations in cocaine concentration on a shorter length scale could be measured with several MSI methods [26, 27]. Matrix-assisted laser desorption/ionisation (MALDI) has been widely applied to study the longitudinal distribution of cocaine in human hair. Porta et al. presented fast screening of 60 mm of hair by MALDI, covering the growth period of approx. half year, with reported spatial resolution of 1 mm [28]. Approximately the same spatial resolution has been reported by Cuypers et al. [29], who studied the effect of hair bleaching on the ability to detect the cocaine in human hair, as well as cocaine reaction products after bleaching. Additional description of MALDI technique on the human hair can be found in a compendium edited by Simona Francese [30].

Longitudinal sectioning of the hair, which reveals its interior, allows the primary ion beam, used for Secondary Ion Mass Spectroscopy (SIMS), to access the hair matrix, thus enabling the detection of chemicals embedded in the hair. In this way, position-sensitive detection of cocaine and its metabolites in hair become feasible, characterized by high lateral resolution of SIMS method. Flinders et al. [31] reported MetA-SIMS analysis of longitudinally cut hair samples, coated by a gold layer to increase chemical sensitivity of conventional SIMS method.

The primary ion impact in SIMS triggers a cascade of billiard-ball-like nuclear collisions and induces dissociation and fragmentation of organic sample material, resulting in low yields of high-mass molecules ejected from the analysed samples, and correspondingly, to a low chemical sensitivity for $m/z > 500$. In a classic SIMS set-up, the absolute yields range from $10^{-4}$ to $10^{-3}$ per single impinging particle. To increase absolute molecular yields and the associated chemical sensitivity, cluster beams were introduced in SIMS, such as $Au_n^+$, $SF_5^+$, $C_{60}^+$, $Bi_3^+$ and $Ar_{(n)}^+$ clusters with n = 55–3000 [32, 33]. More recently, the use of water clusters in SIMS has been demonstrated to improve both aspects even further [34, 35].

Alternatively, atomic heavy ions with energies in the MeV range may be applied in an innovative analytical technique now referred to as MeV-SIMS [36–38]. In contrast to standard SIMS the ion beam interacts with the electrons instead of the nuclei in the sample. Dense energy transfer from the projectile to the electrons along the primary ion track induces shock waves in the target material and results in high probability for desorption of large, non-fragmented secondary molecular ions [36, 39–41], therefore, absolute yields can be as high as $10^{-1}$–$10^{-2}$ per primary ion [41]. Upper limit of detected non-fragmented molecular masses is strongly dependent on the electron stopping power for the applied ion, and may expand to several 10 kDa for very heavy swift ions applied as primary beams, i.e. 90 MeV $^{127}$I. The technique features a high potential lateral resolution; as no sample-surface pre-processing is required and the damage cross-section associated with the impact of an individual primary ion is in the order of few square nanometers [42], the lateral resolution of MeV SIMS is primarily determined by the size of the primary ion beam. Submicrometer lateral resolution studies with MeV-SIMS were recently reported [43, 44]. Our results show that MeV-SIMS allows determining cocaine concectration in human hair with a lateral resolution down to 50 $\mu$m without the need for a chemical preparation of the sample.

## Experimental

Time-Of-Flight (TOF) mass spectrometer for purpose of MeV-SIMS analysis has been implemented at the high-energy focused-ion-beam facility of the Jožef Stefan Institute (JSI) (Fig 1). The ion focusing system equipped with magnetic quadrupole triplet lens is able to focus ion beams in the geometrical centre of the chamber. A motorised, five-axis vacuum goniometer, optical zoom microscope, two X-ray detectors for low and high X-ray energies, a detector for backscattered ions, and an on-off axis STIM detector are integrated together with the TOF spectrometer in the measuring chamber [45]. The TOF spectrometer is installed at the angle of 55˚ in respect to the primary beam direction. The descibed system has since been further upgraded.

A $^{35}$Cl$^-$ beam is extracted from a negative caesium sputtering ion source and accelerated with a 2 MV tandem accelerator. A beam of $^{35}$Cl$^{6+}$ ions is selected among the charge states at the exit of the accelerator by a switching magnet. This primary ion beam is pulsed using a combination of a parallel-plate deflector positioned 9.5 metres from the sample and a collimating slit positioned two metres from the sample [45]. One of the deflector plates is biased at a constant voltage of +900 V, and the second is connected to a fast, high-voltage switch that is able to raise the voltage with a 10 ns risetime to a potential of 900 V and thus bring the beam into a non-deflected (open) position for periods ranging between 20 ns and 230 ns. Data is collected and proccessed using a combination of software supplied by Oxford Microbeams and in-house-written code.

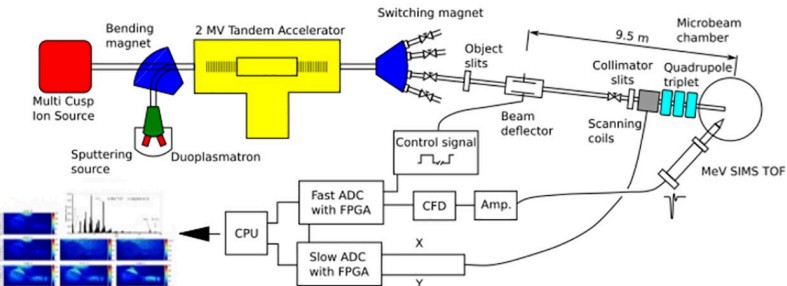

**Fig 1. Schematic view of the MeV-SIMS setup at the JSI tandem accelerator.**

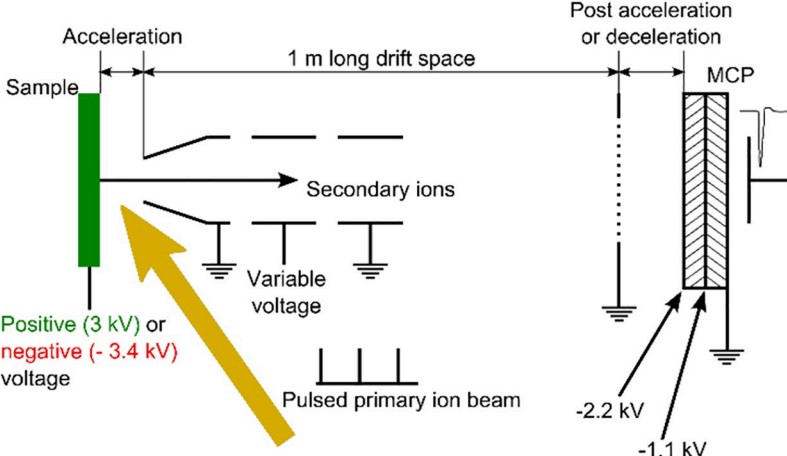

**Fig 2. Operation of a linear TOF mass spectrometer mounted at 55˚ to the beam-line direction.** The secondary ions are extracted by a voltage applied to the sample holder and directed by an Einzel lens toward the microchannel plate detector positioned at the end of a 1-m-long field-free drift region.

The schematic operating principle of the linear Time-Of-Flight (TOF) mass spectrometer is shown in Fig 2. Short pulse of primary 5.8 MeV $^{35}Cl^{6+}$ ions hits the sample. A constant bias voltage of +3 kV applied to the sample accelerates the desorbed secondary ions from the sample towards the grounded extraction electrode of the TOF mass spectrometer. An electrostatic Einzel lens directs the secondary-ion flow towards the microchannel plate (MCP) positioned at the end of a 1 m long field-free drift tube [45]. The surface of the first MCP is biased at a fixed voltage of −2.2 kV and provides an additional acceleration for the positive secondary ions, which increases the efficiency of the MCP detector. By inverting the target bias voltage to negative −3.4 kV, the system allows the operation in negative ion mode, where negatively charged secondary ions are detected.

MeV-SIMS is performed directly at the pristine sample surface. Due to the high yield of secondary ions, extremely low primary ion currents in the fA range can be applied in MeV-SIMS, which diminishes the surface charging of any non-conductive sample materials to an extent where it does not significantly deteriorate the mass spectral quality. Sample coating with dedicated matrices, as is the case with MALDI, or by conductive layers in MetA-SIMS, is not required for MeV-SIMS. The damage cross section, associated with the impact of single primary ion, is in the order of few square nanometers [42]. As a result, the lateral size of the primary ion beam predominantly determines the spatial resolution of MeV-SIMS. The achievable beam size at the high-energy focused ion-beam facility at JSI depends on the brightness of the ion source, and amounts $600 \times 600$ nm$^2$ for protons [46]. For the chlorine ions used for MeV-SIMS, the existing brightness of the sputter ion source is significantly lower, resulting in a 5.8 MeV $^{35}Cl^{6+}$ beam with sufficient ion current for MeV-SIMS with a beam size of approx. $10 \times 10 \, \mu m^2$ in a pulsed beam mode. If the thickness of an analysed sample is less than 10 $\mu$m, then an alternative aproach with a continuous primary ion beam can improve the beam resolution to sub micrometre level. However, such approach is not possible with human hair samples [43].

## Calibration

We deposited three distinct chemical compounds (arginine, leucine and cholesterol) with controlled molecular composition and molecular masses ranging from 174 to 384 on a clean

silicon wafers to compare and contrast the molecular information obtained by standard TOF SIMS technique and by MeV-SIMS. TOF mass spectra were measured by commercially available state-of-the-art *IONTOF TOF.SIMS 5* spectrometer applying 25 keV $^{209}Bi_3^+$ primary ion beam, and by home-built TOF spectrometer for MeV-SIMS applying 5.8 MeV $^{35}Cl^{6+}$ primary ion beam. Spectral resolution $m/\Delta m$ was 300 for MeV-SIMS and $10^4$ for regular SIMS.

Fig 3 shows keV-SIMS and MeV-SIMS mass spectra of arginine, cholesterol and leucine using a 25 keV $Bi^{3+}$ (lower orange spectra) and 5.8 MeV $^{35}Cl^{6+}$ (upper blue spectra) primary ion beams, respectively. The mass resolution $m/\Delta m$ of the home-made linear TOF for

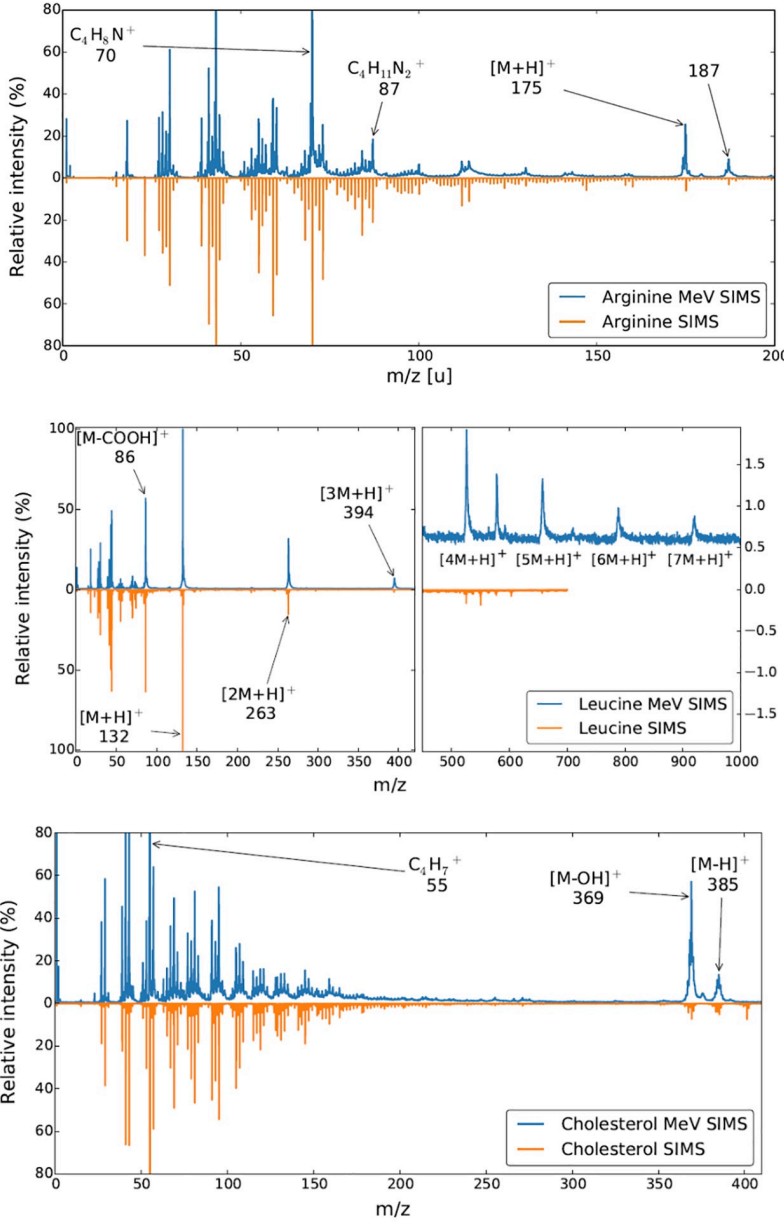

**Fig 3. A qualitative comparison of 5.8 MeV $^{35}Cl^{6+}$ MeV-SIMS (*blue*) and 25 keV $Bi^{3+}$ *IONTOF TOF.SIMS 5* (*orange*) spectra of arginine, cholesterol and leucine used for calibration.** The keV SIMS method does indeed offer better resolution, especially at lower energies, while MeV-SIMS gives higher yields for non-fragmented protonated molecules.

MeV-SIMS method amounts 400 at $m/z$ value of 300, whereas the *IONTOF TOF.SIMS 5* spectrometer mass resolution is one order of magnitude higher at the same $m/z$ value. The mass spectral composition in the low mass region is remarkably similar within the limits of comparison given the different mass resolution of the two instruments. The protonated molecular peak of arginine at an $m/z$ of 175.2 is very pronounced, together with the fragments $[C_4H_8N]^+$, $[CH_5N_3]^+$ and $[CH_3N_2]^+$ with $m/z$ values of 70, 59 and 43, respectively. The cholesterol spectrum features a protonated molecular peak $[M+H]^+$ at 385 $m/z$. In both cases, the normalized yield of non-fragmented protonated parent molecules is much stronger in MeV-SIMS spectra. Normalized yields of MeV-SIMS for the case of leucine and arginine were approx. $3 \times 10^{-2}$ detected $[M+H]^+$ molecular ions per single 5.8 MeV $^{35}Cl^{6+}$ primary ion [41], which is typically 3 orders of magnitude higher than in the comparable keV-SIMS setups with $Bi^{3+}$ clusters. This illustrates the advantage MeV SIMS over regular SIMS introduced above.

## Sample preparation

A set of hair samples, which were previously analyzed by means of MALDI TOF-MS technique, was used to examine the evolution of the cocaine signal by MeV-SIMS along the hair longitudinal cross-section. All of the samples were collected post-mortem and were used previously by Cuypers et al. [29] and Flinders et al. [31]. Within these works, more sets of samples from the same subject were analyzed, and provided similar results within the sets. Approval for this study was received from the Medical Ethics Committee of the faculty of Medicine of the University Hospital of Leuven, Belgium on June 27 2017 and it was registered under the national number NH019-2017-06-01 in Belgium. The first of the samples was from an actual cocaine user, the second one was artificially contaminated (soaked in a cocaine-HCl solution) and the last sample was control [29]. For our purposes the analysed hair was sliced longitudinally using a dedicated tool with a cryotome knife [31], an upgrade of the earlier suggested cutting equipment [47]. The hair sample was positioned inside a machined groove on a metallic block, with its depth individually selected for each hair, varying from 30 to 70 micrometers in 5-micrometer step. A cutting device with an incorporated cryotome knife is designed to accurately slide over the metallic block. A single pass of the cutting device over the hair sample positioned in a groove results in an evenly opened longitudinal half-cross-section of the hair sample with a length of 4 cm. The processed samples were stretched over a silicon wafer and attached at the end sections by a metallic tape, facing the opened flat surface of hair shaft upwards to enable the access of the probing primary ion beam. The mounted samples are shown in Fig 4.

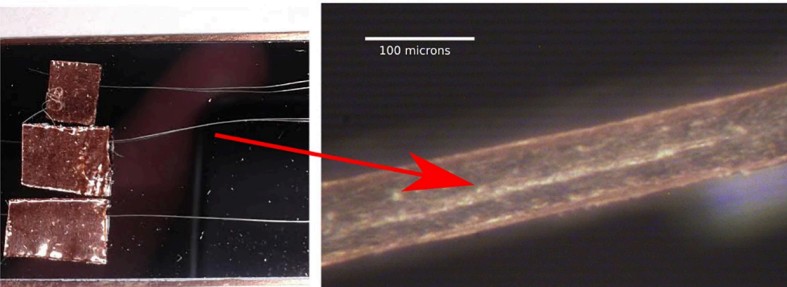

**Fig 4.** *Left*: Hair samples attached to a silicon wafer. *Right*: One of the samples as seen under the microscope.

## Measurements

Hair samples were raster-scanned with the primary beam, using magnetic beam deflection. The basic scanning frame consists of $256 \times 256$ measuring points, yielding a square shaped scanning field in the case of perpendicular beam impact. For MeV-SIMS, the sample holder is tilted 55 degrees with regards to the primary beam axis in order to be positioned perpendicularly to the axis of the TOF spectrometer. The individual scanning frame is consequently rectangular, with horizontal dimension equal to $1/\cos55°$ of the vertical one. In our case, the dimension of the individual scanning frame was selected to be $250 \times 436\ \mu m^2$.

# Results and discussion

The spectra from one of the measurements of the longitudinally cut hair samples are shown in Fig 5. The ion beam was focused on the inside of the hair and did not probe the edges. The exact mass of cocaine at m/z = 304.1540 has before been measured by MALDI MS/MS on another set of samples (see Ref. [31]). In contrast with comparable MetA-SIMS measurements, no other significant mass peaks were observed in the $m/z$ range between 200 and 350. Both the spectra obtained for the control sample and for the normal hair sample dipped in cocaine solution show absence of the cocaine peak at m/z = 304. The hair sample dipped in a saturated cocaine water solution and air-dried show no signs of cocaine diffusion or permeability even in the outer rim of the hair. This might be due to the primary beam probing of the interior, not

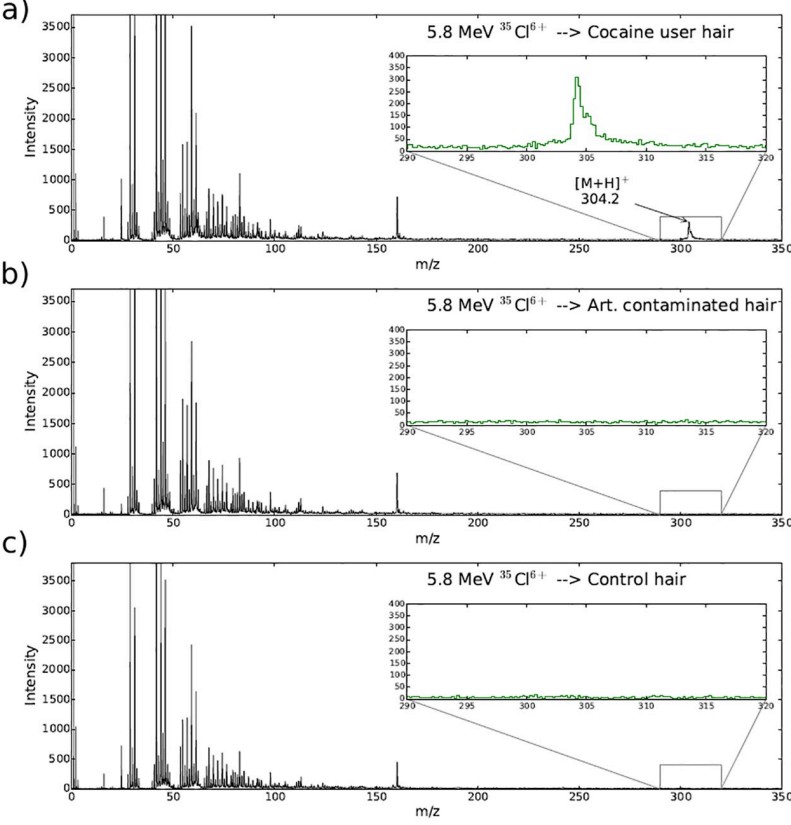

**Fig 5. Measured MeV-SIMS spectra of three hair samples.** The cocaine peak was measured only in the sample from the actual cocaine user, while the spectra of the artificially contaminated hair exhibits no such peak as the ion beam was focused on the inside of the hair samples.

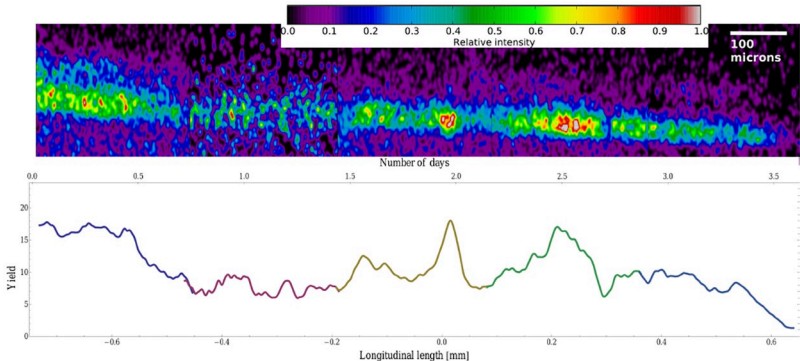

**Fig 6.** *Top*: The distribution of the cocaine signal along the user's hair sample. *Bottom*: Laterally integrated cocaine intensity along the hair sample. Different colours represent different peaks or regions of interest in the spectrum. As explained in the text, the time scale was approximated with the average hair-growth rate of 0.4 mm/day.

the exterior of the hair due to the sample sectioning and mounting geometry, where the contaminated surface of the hair is not directly exposed to the analytical beam. This result indicates high sensitivity for cocaine metabolically incorporated in the interior hair structure, which is not affected by external contamination of the hair.

We measured a series of rectangular frames along the hair of the cocaine user, each with lateral dimensions of $250 \times 400 \ \mu m^2$ to obtain information on the longitudinal variations in the concentration of cocaine. To be able to compare the relative yields of the peak at $m/z = 304$ in the sequence of frames, an overlapping region of 20 micrometres was used during the acquisition of consecutive frames to renormalize the yields in the individual scans. The resulting image and the integrated $m/z$ 304 intensity distribution along the hair are shown in Fig 6. Three regions of corresponding elevated cocaine intensities were observed. The first broad region corresponds to the longitudinal scale in Fig 6 from −0.8 mm to −0.55 mm. The other two peaks at 0.02 mm and 0.21 mm centroid positions are significantly narrower.

An approximate time scale could be estimated with an assumed average hair-growth rate [6] of 0.4 mm/day. Based on this scale, the two peaks at the 0.02 mm and 0.21 mm positions are approximately 12 hours apart. The pronounced peak in the cocaine concentration at the 0.02 mm position spans over a longitudinal distance of 50 micrometres. Converted to time scale, this distance approximately corresponds to a hair-growth period of 3 hours suggesting a temporal regime of cocaine intake.

## Conclusion

MeV-SIMS combines high chemical sensitivity and good lateral resolution, a combination appropriate to measure the distribution of the pharmaceutical compounds and drugs incorporated in the human hair. In the reported case of cocaine detection in human hair, we observe pronounced longitudinal spikes of cocaine's relative concentration. Results indicate highly spatially structured incorporation of cocaine in hair, that are indicative of rapid fluctuations of its body concentrations. Due to the highly focused $10 \times 10 \ \mu m^2$ primary beam coupled with precise scanning across the sample we were able to distinguish peaks as narrow as 50 $\mu$m. These correspond to a temporal resolution on an hourly level, allowing us to reconstruct a detailed history of cocaine intake. Additionally, exposing the sample core by cutting the sample longitudinally nullifies any false signals due external contamination of an otherwise 'clean' sample.

The specific desorption and ionization process of sample material in MeV-SIMS is based on energy deposition in the sample material via electronic stopping. This introduces specific advantages in comparison with other comparable secondary ion MS techniques: a higher yield of non-fragmented secondary molecular ions per primary ion impact. This enables surface analysis in a strictly static regime directly on intact biological materials, where the sample is neither coated nor significantly chemically altered. In turn, this facilitates subsequent additional chemical analysis at the same surface (volume) region of the sample which shows how the technique could be used complementarily with more standard forensic methods.

## Author Contributions

**Conceptualization:** Ron M. A. Heeren.

**Formal analysis:** Luka Jeromel, Nina Ogrinc.

**Investigation:** Luka Jeromel, Nina Ogrinc, Boštjan Jenčič.

**Methodology:** Luka Jeromel, Katarina Vogel-Mikuš, Bryn Flinders, Eva Cuypers.

**Resources:** Zdravko Siketić, Primož Vavpetič, Mitja Kelemen, Janez Kovač.

**Software:** Zdravko Rupnik, Klemen Bučar, Matjaž Vencelj.

**Supervision:** Primož Pelicon.

**Writing – original draft:** Luka Jeromel.

**Writing – review & editing:** Žiga Barba.

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
