## [Decision Letter · Decision Letter 0]

7 Dec 2021

PONE-D-21-29597Molecular imaging of humain hair with MeV-SIMS: a case study of cocaine detection and distribution in the hair of a cocaine userPLOS ONE

Dear Dr. Barba,

Thank you for submitting your manuscript to PLOS ONE. After careful consideration, we feel that it has merit but does not fully meet PLOS ONE’s publication criteria as it currently stands. Therefore, we invite you to submit a revised version of the manuscript that addresses the points raised during the review process.

Please include the following items when submitting your revised manuscript:A rebuttal letter that responds to each point raised by the academic editor and reviewer(s). You should upload this letter as a separate file labeled 'Response to Reviewers'.A marked-up copy of your manuscript that highlights changes made to the original version. You should upload this as a separate file labeled 'Revised Manuscript with Track Changes'.An unmarked version of your revised paper without tracked changes. You should upload this as a separate file labeled 'Manuscript'.

We look forward to receiving your revised manuscript.

Kind regards,

Joseph Banoub, Ph,D., D. Sc.

Academic Editor

PLOS ONE

Journal Requirements:

Work at JSI was supported by the Slovenian research agency grants No. P1-0112, I0-0005, J7-9398 and N1-0090. Additionally, resources and within the EU H2020 project No. 824096 “RADIATE ” and IAEA CRP projects F11019 “Development of Molecular Concentration Mapping Techniques Using MeV Focused Ion Beams" and 

F11021 ”Enhancing Nuclear Analytical Techniques to Meet the Needs of Forensic Science”. N.O. 

acknowledges funding by the European Union, European Social Fund, and the support 

from FP7 European Union Marie Curie IAPP Program, BRAINPATH. A traveling 

grant for a long stay abroad of E.C was awarded by Fonds Wetenschappelijk Onderzoek 

(FWO).

Work at JSI was supported by the Slovenian research agency grants No. P1-0112, 231

I0-0005, J7-9398 and N1-0090. Additionally, resources and within the EU H2020 project 232

No. 824096 “RADIATE” and IAEA CRP projects F11019 “Development of Molecular 233

Concentration Mapping Techniques Using MeV Focused Ion Beams” and F11021 234

”Enhancing Nuclear Analytical Techniques to Meet the Needs of Forensic Science”. N.O. 235

acknowledges funding by the European Union, European Social Fund, and the support 236

from FP7 European Union Marie Curie IAPP Program, BRAINPATH. A traveling 237

grant for a long stay abroad of E.C was awarded by Fonds Wetenschappelijk Onderzoek 238

(FWO). 

Work at JSI was supported by the Slovenian research agency grants No. P1-0112, I0-0005, J7-9398 and N1-0090. Additionally, resources and within the EU H2020 project No. 824096 “RADIATE ” and IAEA CRP projects F11019 “Development of Molecular Concentration Mapping Techniques Using MeV Focused Ion Beams" and 

F11021 ”Enhancing Nuclear Analytical Techniques to Meet the Needs of Forensic Science”. N.O. 

acknowledges funding by the European Union, European Social Fund, and the support 

from FP7 European Union Marie Curie IAPP Program, BRAINPATH. A traveling 

grant for a long stay abroad of E.C was awarded by Fonds Wetenschappelijk Onderzoek 

(FWO).

Additional Editor Comments:

I have added the second referee comments below.

Please try to answer the comments of referee #1 when you resubmit this manuscript.

Referee #2 comments

The paper by Jeromel,L. et al looks at distribution of cocaine in human hair, using TOF-SIMS. The paper is well written and edited, and is recommended for publications. However the authors may clarify a few points listed below in the manuscript. These are suggestions only.

Somewhere in the introduction, the question regarding the standards such as cholesterol or the specific amino acids should be clarified. Why were these specific standards used, considering cholesterol is hydrophobic compared to the amino acids. Do these ions have any specific desorption rates compared to cocaine? Was free cocaine in solution form studied?

I am not sure how changes occur in “post mortem” state of the hair cells? Kindly clarify.

In the last figure the zones of high concentration (red) of cocaine at specific spots of the hair is due to especial location of the hair due to any structures of the hair? Why are the signals stronger from these specific areas? Kindly describe the figure in more details.

Reviewers' comments:

Reviewer's Responses to Questions

**Comments to the Author**

1. Is the manuscript technically sound, and do the data support the conclusions?

Reviewer #1: Partly

2. Has the statistical analysis been performed appropriately and rigorously? 

Reviewer #1: No

3. Have the authors made all data underlying the findings in their manuscript fully available?

Reviewer #1: Yes

4. Is the manuscript presented in an intelligible fashion and written in standard English?

Reviewer #1: Yes

5. Review Comments to the Author

Reviewer #1: In the manuscript “Molecular imaging of humain hair with MeV-SIMS: a case study of cocaine detection and distribution in the hair of a cocaine user”, Jeroma et al. test a methodology for examining cocaine distribution within human hair. For this reviewer, it took a little time to discern whether the authors were presenting a research article or a methodology article. Regardless, the examination of cocaine in hair is far from novel, and there are a number of methodologies for this purpose, including mass spectrometry based; the authors did provide a sufficient introduction to cover some of these methodologies. Specific comments are as follows:

1. The authors need to examine the same hair more than once, to show the variability of the method.

2. More than one hair sample should be examined, to ensure comparable spectra can be obtained from cocaine users.

3. Please provide the ms/ms spectra for m/z 304, to verify to a reader that the ion is indeed cocaine.

4. Can the cocaine be quantified (rather than providing relative intensity information) using this method? This would be quite important.

5. Some modest edits to language (including within the title) are needed.

6. PLOS authors have the option to publish the peer review history of their article (what does this mean?). If published, this will include your full peer review and any attached files.

Reviewer #1: No

---

## [Author Response · Author response to Decision Letter 0]

13 Jan 2022

List of changes made based on the remarks of the Reviewer 1:

Technical questions Reviewer #1

“The authors need to examine the same hair more than once, to show the variability of the method.”

We have analyzed the sample more than once, and the measurements have been done over a long period of time (several months). Results remained the same, however the ion yield of molecules, such as cocaine, has been reduced when the samples aged due to the oxidation of the samples. We revised the text accordingly so it is now clear that the measurements were performed multiple times

“More than one hair sample should be examined, to ensure comparable spectra can be obtained from cocaine users.”

Due to the strict ethical clearance policy, we only obtained one sample. However, multiple samples of the same subject were analyzed within the work of Flinders (see ref 31), and later one of these samples was forwarded to Jožef Stefan Institute. The manuscript was changed accordingly.

“Please provide the ms/ms spectra for m/z 304, to verify to a reader that the ion is indeed cocaine.”

Samples of the same subject were analyzed with MALDI MS/MS mass spectrometry imaging, where the exact mass of cocaine is measurable. See ref 31. After MeV-SIMS analysis, the same set of samples was also measured by DAPNe analysis, where Orbi trap mass spectrometer also confirmed the presence of cocaine molecular peak. The text was changed accordingly. 

“Can the cocaine be quantified (rather than providing relative intensity information) using this method? This would be quite important.”

Quantification with MeV-SIMS, as well as common SIMS, is not possible due to the high impact the surrounding matrix on the ionization probability. A standardization of secondary ion yield of the molecule in relation to the concentration, given the constant matrix (e.g. collagen in human hair) would be possible, however even such approach would lead to questionable results due to uneven composition of biological tissues. Our scope remained only on providing relative intensity information. 

“Some modest edits to language (including within the title) are needed.”

We have revised the text again.

Technical questions Reviewer #2

“Somewhere in the introduction, the question regarding the standards such as cholesterol or the specific amino acids should be clarified. Why were these specific standards used, considering cholesterol is hydrophobic compared to the amino acids. Do these ions have any specific desorption rates compared to cocaine? Was free cocaine in solution form studied?”

Amino-acids, such as arginine, leucine, phenylalanine, etc. as well as cholesterol are, to our knowledge, commonly used in SIMS community as a reference/calibration standards. Therefore, we could draw some explicit comparisons between sputtering rate of MeV and keV ions, not only from the keV-SIMS measurements done at JSI, but also worldwide, and the results encouraged us to develop the method further. The masses of such amino acids and cholesterol are also within the same domain as the mass of cocaine, so such standards provided us the information about expected molecular yields from human hair.

“I am not sure how changes occur in “post mortem” state of the hair cells? Kindly clarify.”

We measured the samples several time over a significant period of time (several months), so we gained some insight into aging of the samples. The localization of cocaine and other hair-specific molecules remained the same, at least on the above cellular size level. However, the intensity of the signal was rapidly diminishing, due to deposition of oxidation layers on the sample. 

“In the last figure the zones of high concentration (red) of cocaine at specific spots of the hair is due to especial location of the hair due to any structures of the hair? Why are the signals stronger from these specific areas? Kindly describe the figure in more details.”

The concentration distribution of cocaine is highly unique compared to distribution of other hair – specific ions. Latter are distributed very homogeneously, which indicates, that there are no significant structural impacts on the secondary ion yield. Indeed, this can greatly influence the results of surface sensitive methods, therefore we were very observant on the intensities of the total ion yield, as well as some other peaks. Among those, the cocaine peak is the only one showing uneven distribution. As it can be seen in the last figure, the gap between the intensiy peaks is speculated to be due to the detection of each individual dose of cocaine that the subject has taken. Through this conclusion, we speculate, that the second dose was taken 12 hours after the first one, and in both cases the cocaine was depositing on the hair for approx. 3 hours. 

Editorial changes

Technical

-Page 5, line 150, sample preparation, technical: “A set of hair samples previouisly analysed using the MALDI TOF-MS technique was used to examine the evolution of the cocaine signal by MeV-SIMS along the hair longitudinal cross-section. All of the samples were collected post-mortem and were used previously by Cuypers et al. [29] and Flinders et al. [31].” 

has been changed to: 

“A set of hair samples, which were previously analyzed by means of MALDI TOF-MS technique, was used to examine the evolution of the cocaine signal by MeV-SIMS along the hair longitudinal cross-section. All of the samples were collected post-mortem and were used previously by Cuypers et al. [29] and Flinders et al. [31]. Within these works, more sets of samples from the same subject were analyzed, and provided similar results within the sets”

-Page 5, line 178, Results and discussion, technical: “The spectra obtained from the surface of the longitudinally cut hair samples are shown in Fig. 5. “, 

has been changed to: 

“The spectra from one of the measurements of the longitudinally cut hair samples are shown in Fig. 5.”

-Page 5, line 181, Results and discussion, technical: 

After

 “The molecular peak of cocaine at m/z of 304 was detected only in the hair 181 sample of the drug user.” 

we added: 

“The exact mass of cocaine at m/z=304.1540 has before been measured by MALDI MS/MS on another set of samples (see ref 31)”

Language

- Page 1, abstract, language: 

“During the growth of human hair, several biomedical substances present in the human body are incorporated into the hair” 

has been changed to: 

“Human hair absorbs numerous biomolecules from the body during its growth”

-Page 1, abstract, language: 

“.It could be either assessed by chemical extraction and analysis of hair bundels, or by direct single hair fibre analysis by mass spectroscopy imaging (MSI).” 

has been changed to: 

“It could be either assessed by chemical extraction and further analysis of hair bundels, or by direct single hair fibre analysis with mass spectroscopy imaging (MSI).”

-Page 1, abstract, language: 

“In this work, we analyzed the cocaine distribution in individual hair samples using MeV-SIMS.” 

has been changed to: 

“Within this work, we analyzed the cocaine distribution in individual hair samples using MeV-SIMS.”

-Page 1, line 1, Introduction, language: 

“The presence of illegal chemical substances in human body is usually monitored by bulk chemical analysis or spectroscopic methods of biological specimens [1–5] such as urine, saliva, sweat, and hair” 

has been changed to: 

“The presence of illegal chemical substances in human body is usually monitored by bulk chemical analysis, or by spectroscopic methods of biological specimens [1–5], such as urine, saliva, sweat, and hair”

-Page 2, line 9, Introduction, language: 

“A number of pharmaceutical substances, as well as illegal substances were detected in hair [9, 10] with a variety of analytical techniques.” 

has been changed to: 

“Various pharmaceutical substances, as well as illegal substances were detected in hair [9, 10] with numerous analytical techniques.”

-Page 2, line 26, Introduction, language: 

“Typical cocaine concentrations found in hair samples after its administration range from 0.5 to 216 ng/mg [23].” 

has been changed to: 

“Typical cocaine concentration values in hair samples after its administration have been reported from 0.5 to 216 ng/mg [23].”

-Page 2, line 42, Introduction, language: 

“Longitudinal sectioning of hair, which reveals its interior, allows the probing primary ion beam with energies of a few keV used in Secondary Ion Mass Spectroscopy (SIMS) to access the hair matrix and enables the detection of chemicals embedded in the hair.” 

has been changed to: 

“Longitudinal sectioning of the hair, which reveals its interior, allows the primary ion beam, used for Secondary Ion Mass Spectroscopy (SIMS), to access the hair matrix, thus enabling the detection of chemicals embedded in the hair.”

-Page 2, line 52, Introduction, language:

 “In a classic SIMS set-up, absolute yields range from 10−4 to 10−3 per single impinging particle. To increase the absolute molecular yields and the associated chemical sensitivity, cluster beams are introduced in SIMS, such as Au+ n , SF+ 5 , C+ 60, Bi+ 3 and Ar+ (n) clusters with n= 55 60–3000 [32, 33] and, more recently, the use of water clusters in SIMS improves both aspects even further [34, 35].” 

has been changed to: 

“In a classic SIMS set-up, the absolute yields range from 10−4 to 10−3 per single impinging particle. To increase absolute molecular yields and the associated chemical sensitivity, cluster beams were introduced in SIMS, such as Au+ n , SF+ 5 , C+ 60, Bi+ 3 and Ar+ (n) clusters with n= 55 60–3000 [32, 33]. More recently, the use of water clusters in SIMS has been demonstrated to improve both aspects even further [34, 35].”

-Page 3, line 76, Experimental, language: 

“Time-Of-Flight (TOF) mass spectrometer for MeV-SIMS is installed at the high-energy focused-ion-beam facility of the Jožef Stefan Institute (JSI) (Fig.1).” 

has been changed to: 

“Time-Of-Flight (TOF) mass spectrometer for purpose of MeV-SIMS analysis has been implemented at the high-energy focused-ion-beam facility of the Jožef Stefan Institute (JSI) (Fig.1).”

-Page 5, line 173, Measurements, language: 

“For MeV-SIMS, the sample holder is tilted for 55 degrees in order to be positioned perpendicularly to the axis of the TOF spectrometer.”

 has been changed to: 

“For MeV-SIMS, the sample holder is tilted 55 degrees with regards to the primary beam axis in order to be positioned perpendicularly to the axis of the TOF spectrometer.”

-Page 6, line 190, Results and discussion, language: 

“This result indicates the method’s high sensitivity for cocaine metabolically incorporated in the interior hair structure and its insensitivity to eventual external contamination of the hair.” 

has been changed to: 

“This result indicates high sensitivity for cocaine metabolically incorporated in the interior hair structure, which is not affected by external contamination of the hair.”

-Page 6, line 212, Conclusion, language: 

“In the reported case of cocaine detection in human hair, we observe pronounced longitudinal spikes of the cocaine levels.” 

has been changed to: 

“In the reported case of cocaine detection in human hair, we observe pronounced longitudinal spikes of cocaine’s relative concentration.”

---

## [Editor Report · Decision Letter 1]

18 Jan 2022

Molecular imaging of human hair with MeV-SIMS: a case study of cocaine detection and distribution in the hair of a cocaine user

PONE-D-21-29597R1

Dear Dr. Barba,

We’re pleased to inform you that your manuscript has been judged scientifically suitable for publication and will be formally accepted for publication once it meets all outstanding technical requirements.

Kind regards,

Joseph Banoub, Ph,D., D. Sc., FCIC, FRCS

Academic Editor

PLOS ONE
---

## [Editor Report · Acceptance letter]

17 Mar 2022

PONE-D-21-29597R1 

Molecular imaging of humain hair with MeV-SIMS: a case study of cocaine detection and distribution in the hair of a cocaine user 

Dear Dr. Barba:

I'm pleased to inform you that your manuscript has been deemed suitable for publication in PLOS ONE. Congratulations! Your manuscript is now with our production department. 

Kind regards, 

on behalf of

Dr. Joseph Banoub 

Academic Editor

PLOS ONE